# Fine-Tuning of the Optical and Electrochemical Properties of Ruthenium(II) Complexes with 2-Arylbenzimidazoles and 4,4′-Dimethoxycarbonyl-2,2′-bipyridine

**DOI:** 10.3390/molecules28186541

**Published:** 2023-09-09

**Authors:** Maria A. Lavrova, Stepan A. Verzun, Sergey A. Mishurinskiy, Maxim A. Sirotin, Sofya K. Bykova, Victoria E. Gontcharenko, Sofia S. Mariasina, Vladislav M. Korshunov, Ilya V. Taydakov, Yury A. Belousov, Vladimir D. Dolzhenko

**Affiliations:** 1Chemistry Department, M.V. Lomonosov Moscow State University, Leninskie Gory Street, Building 1/3, 119234 Moscow, Russia; stepan.verzun@chemistry.msu.ru (S.A.V.); sergei.mishurinskii@chemistry.msu.ru (S.A.M.); maksim.sirotin@chemistry.msu.ru (M.A.S.); sofia.mariasina@yandex.ru (S.S.M.); belousov@inorg.chem.msu.ru (Y.A.B.); 2N.N. Semenov Federal Research Center for Chemical Physics, Kosygina Street 4, 119991 Moscow, Russia; 3Higher Chemical College of RAS, Mendeleev University of Chemical Technology, Miusskaya Square, 9, 125047 Moscow, Russia; s.k.bykova@gmail.com; 4Faculty of Chemistry, National Research University Higher School of Economics, 20 Miasnitskaya Street, 101000 Moscow, Russia; victo.goncharenko@gmail.com; 5P.N. Lebedev Physical Institute of the Russian Academy of Sciences, 119991 Moscow, Russia; korshunovvm@lebedev.ru (V.M.K.); taydakov@gmail.com (I.V.T.); 6Faculty of Fundamental Medicine, Lomonosov Moscow State University, 119991 Moscow, Russia; 7Institute of Functional Genomics, Lomonosov Moscow State University, 119992 Moscow, Russia; 8Faculty of Fundamental Sciences, Bauman Moscow State Technical University, 105005 Moscow, Russia; 9Academic Department of Innovational Materials and Technologies Chemistry, G.V. Plekhanov Russian University of Economics, 36 Stremyannoy per., 117997 Moscow, Russia; 10N.D. Zelinsky Institute of Organic Chemistry, Russian Academy of Sciences, Leninsky pr. 47, 119991 Moscow, Russia

**Keywords:** cyclometalated complexes, photosensitizers, thiocyanate-free ruthenium dyes, bipyridine ruthenium complexes, benzimidazoles

## Abstract

A series of cyclometalated complexes of ruthenium (II) with four different substituents in the aryl fragment of benzimidazole was synthesized in order to study the effect of substituent donation on the electronic structure of the substances. The resulting complexes were studied using X-ray diffraction, NMR spectroscopy, MALDI mass spectrometry, electron absorption spectroscopy, luminescence spectroscopy, and cyclic voltammetry as well as DFT/TDDFT was also used to interpret the results. All the complexes have intense absorption in the range of up to 700 nm, the triplet nature of the excited state was confirmed by measurement of luminescence decay. With an increase in substituent donation, a red shift of the absorption and emission bands occurs, and the lifetime of the excited state and the redox potential of the complex decrease. The combination of these properties shows that the complexes are excellent dyes and can be used as photosensitizers.

## 1. Introduction

Ruthenium (II) polypyridine complexes have a wide range of applications: they are used as photocatalysts [1], luminescent chemosensors [2], dyes in Grätzel cells [3,4,5], anticancer drugs [6], etc. The main feature of these complexes is the presence of intense absorption in the visible region, which makes them very promising for usage in light conversion molecular devices. Classic complexes for this purpose are isothiocyanate complexes with monodentate NCS ligands [7]. However, such architecture has poor stability, due to which they are inappropriate for use in solutions. As an alternative bidentate and tridentate N-donor ligands based on pyridine and pyrazole rings are used instead of isothiocyanate ligands [8,9]. Another way to increase the stability of complexes is cyclometalation, i.e., the formation between the metal and carbon atoms from the ligand. Here and below, cyclometalated ligands will be referred to as C^N ligands, since they are coordinated to the ruthenium atom by carbon and nitrogen, and N-donor bidentate ligands as N^N ligands. It is important to note that when a cyclometalated fragment is introduced into the complex, photophysical and electrochemical properties of the latter dramatically change [8]. Therefore, the analysis of factors affecting the properties of complexes is an urgent task. One approach to the search for new dyes is to vary the donor-acceptor nature of substituents in C^N ligands. The highest occupied molecular orbital (HOMO) is localized on the d-orbitals of ruthenium and the orbitals of the C^N ligand [8,10,11]. The lowest unoccupied molecular orbital (LUMO) is localized mainly on the N^N ligand. Therefore, the change in the electronic structure of the ligand is directly related to the HOMO-LUMO gap, and, as a result, to the position of the absorption edge, the emission maximum, and the redox potential. The most studied cyclometalated complexes are the ones bearing phenylpyridines, phenylazoles, or terpyridines [7,12,13,14,15,16,17,18,19,20,21,22,23,24,25].

Nevertheless, phenylpyridines are complex objects for synthesis, and there are no papers where strong electron-donor substituents have been introduced into them, although the effect of electron-withdrawing substituents has been extensively studied [14,15,16]. As an alternative class of ligands, benzimidazoles can be used. These compounds are convenient objects for precise changes in the electronic structure since they are readily available synthetically, and it is possible to introduce a wide range of substituents at different positions of the aryl and benzimidazole fragments.

Several ruthenium compounds with similar ligands with an imidazole moiety have been described in the literature. The first type (Figure 1a) is N^C^N ligands containing two benzimidazole fragments, such as 2,6-bis(benzimidazol-2-yl)benzene [26,27,28,29,30,31,32,33,34,35,36,37]. The second type is C^N ligands (Figure 1b), where the Ru-C bond is formed with the imidazole fragment and one of the imidazole nitrogen is bonded to a pyridine [27,38]. The third type is the N^N ligands of the 2-benzimidazol-2-yl-pyridines class (Figure 1c) [27,39,40,41,42,43,44,45,46,47]. The C^N ligands based on 2-arylimidazoles that interest us are presented in Figure 1d [48,49,50].

Ligands of the first type (Figure 1a) are interesting for the design of ruthenium binuclear complexes, and electron transfer from one metal center to another is mainly investigated for them [26,32,33,37]. The modification of such ligands is mainly done by carboxyl or ester groups or phenyl groups [26,32]. Such complexes can be used for photodynamic therapy [31], photooxidation of water [34], and electropolymerization [35].

Complexes with ligands of the second type (Figure 1b) have also been studied as sensitizers for DSSC [38]. For example, the authors of [38] studied the effect of the size of the conjugated system of C^N-ligand on the photosensitizer efficiency. 

Ligands of the third type (Figure 1c) are N^N-donors, i.e., complexes with them are not cyclometalated. Complexes for cancer immunotherapy have been obtained with such pyridine-containing ligands [47], and sensitizers for DSSCs have been investigated [42,43,44,45,46].

Finally, there are several studies where complexes with ligands of the fourth type (Figure 1d) have been investigated [48,49,50]. For example, in [49], ruthenium complexes with 2-arylbenzimidazoles modified with a carboxyl group were investigated as anticancer drugs. In [50], cyclometalated ruthenium complexes with unsubstituted 2-phenylimidazole and various N^N-donor ligands were proposed as photosensitizers. The closest to the topic of this work is the study of cyclometalated ruthenium complexes with 1-benzyl-2-aryl-benzimidazoles, which showed good performance in DSSC, where -CF_3_ groups and N-hexylphenothiazine [48], i.e., only acceptor substituents, were introduced into the aryl moiety. The introduction of the strong acceptor N-hexylphenothiazine shifts the absorption bands to the red region, but the cell sensitized by it shows lower efficiency relative to the sensitizer with -CF_3_ groups. Cyclometalated iron complexes have also been studied with similar ligands as sensitizers for DSSCs [51]. In addition to the ligands used in [48], the authors introduced a donor-NMe_2_ group into the aryl fragment, which led to the desired destabilization of the MC-state.

In addition to ruthenium(II) complexes, benzimidazoles have been used as C^N ligands in iridium(III) complexes [52]. It is shown that the expansion of the conjugated ligand system and its geometry have a significant effect not only on the electronic structure of the complex, but also on its structure and composition [53,54,55].

Previously, we studied cyclometalated complexes of ruthenium(II) [56] and iridium(III) [57] with 1-phenyl-2-aryl-benzimidazoles and showed that the benzimidazole and aryl parts of the ligand strongly differ from each other in donor–acceptor properties, which leads to the prevalence of ligand-to-ligand charge transfer (LLCT) in molecules. As a way to solve this problem, we propose to introduce an electron-donor substituent (e.g., methyl) into the benzimidazole fragment. Methyl radical was chosen because it is the simplest sigma electron-donating group. Then, 4,4′-dimethoxycarbonyl-2,2′-bipyridine (dmdcbp) was chosen as the N^N ligand. Usually, 4,4′-dicarboxy-2,2′-bipyridine (dcbp) is used as an anchoring ligand. We use it not as an acid but as an ester, as this increases the solubility of the complexes and facilitates their study. In our previous work [56], it was shown that the hydrolysis is easily carried out in one step and that the optical properties of the complex with dcbp and dmdcbp do not differ practically.

In this work, we have designed and synthesized a series of cyclometalated Ru(II) complexes with 2-aryl-5-methylbenzimidazoles bearing electron-withdrawing (NO_2_) and electron-donating substituents (OMe, NMe_2_) and anchoring 4,4′-dimethoxycarbonyl-2,2′-bipyridine. We studied the composition and structure of complexes with NMR spectroscopy, MALDI mass spectrometry, and X-ray diffraction. Photophysical properties were studied with UV-vis and luminescence spectroscopy, redox potentials were measured by use of cyclic voltammetry (CV). These results were interpreted together with their electronic structure, obtained by a combined DFT/TD-DFT approach. The results were compared to our previous work [56].

## 2. Results

### 2.1. Synthesis and Characterization

C^N ligands were prepared via 3-step synthesis from 4-methyl-2-nitroaniline (Figure 1). The ligands were obtained with a high yield by condensation of N-benzyl-4-methyl-o-phenylenediamine and bisulfite adducts of corresponding aldehydes with various substituents using standard procedure [58]. The desired diamine was produced by the reduction of 4-methyl-2-nitro-N-benzylaniline with hydrazine using Raney nickel [59]. The precursor for reduction was obtained by benzylation of 4-methyl-2-nitroaniline with benzylchloride (BnCl) [60]. This route of synthesis was chosen in order to avoid isomer formation.

Complexes **1**–**4** were prepared via a standard [56] 2-stage method (Figure 2). In the first stage, cyclometalation was carried out. The product is unstable in solution; therefore, it was used in the second stage without characterization, and the composition of the complex was assigned in accordance with the literature [25]. The solid contained some amount of p-cymene, which was not evaporated under reduced pressure. That is why the shortage of dmdcbp was taken on the 2nd stage—it is hard to separate the final product from the unreacted dmdcbp. It is interesting to note that an increase in the acceptor properties of a substituent leads to a decrease in the yield of cyclometalation. In the second stage, the dimethyl ester of dicarboxybipyridine (dmdcbp) was introduced into the complex. We decided to use ester instead of acid in order to simplify the isolation and purification of complexes. The resulting compounds were characterized by ^1^H NMR, HRMS, complexes **1** and **3** were studied by X-ray crystallography.

#### 2.1.1. NMR Spectroscopy

The composition and purity of the complexes were verified by ^1^H NMR spectroscopy. The spectrum of the free C^N ligand is very different from that of the cyclometalated one. First, one of the protons of the aryl fragment disappears as a result of the formation of a ruthenium–carbon covalent bond. This also causes the signals of neighboring protons to change. Second, the formation of a covalent bond with a metal leads to a redistribution of the electron density in the ligand and, as a consequence, to a strong change in the chemical shift of the ligand proton signals. The proton signals of the free N^N ligand also differ greatly from the coordinated one (Figure 2). It is interesting to note that bipyridine rings become non-equivalent when coordinated with a metal. This is clearly seen in signals in the region of 8.9–9.1 ppm. Protons from four different bipyridyl rings have different chemical shifts due to different Ru-N bond lengths. One of the signals is shifted more than the others because its proton is located in the pyridyl ring, the nitrogen of which is farthest from the ruthenium (Table 1).

#### 2.1.2. Crystal Structures

Single crystals of compound **1** were obtained from CH_2_Cl_2_:CHCl_3_ 1:1 mixture under slow solvent evaporation; single crystals of complex **3** suitable for X-ray diffraction study were obtained from CH_2_Cl_2_:hexane 3:1 mixture.

The asymmetric unit of crystal structure of **1** contains C^N cyclometalated [Ru(L-NO_2_)(dmdcbp)_2_]^+^ complex cation, PF_6_^−^ anion and 5 solvated chloroform molecules, while the asymmetric unit of crystal structure of **3** consists of C^N cyclometalated [Ru(L-(OMe)_2_)(dmdcbp)_2_]^+^ complex cation, PF_6_^−^ anion, and solvated hexane molecule (Figure 3). In both structures, the ruthenium atom is located in a slightly distorted octahedral coordination environment and is coordinated by two nitrogen atoms of each of the bipyridine ligands and one nitrogen and one carbon atom of the C^N ligand; the list of selective bonds is presented in Table 1. 

It is to be noted that the Ru1-N6 bond trans to Ru1-C1 is elongated compared to other Ru-N bonds, which can be explained by the trans effect [61]. According to the analysis of the distribution of Ru-C bond lengths in C^N cyclometalated complexes published in the Cambridge Structure Database (CSD version 5.44, update June 2023), Ru1-C1 bond length is typical. Analysis of the crystal packing of **1** (Figure 4) is additionally stabilized by weak intermolecular interactions between the carbon atom of solvated chloroform and the carboxyl oxygen atom of the dimethyl ester of dicarboxybipyridine (C1S-H1S…O3, C1S…O3 distance is 3.14 Å). In the meantime, the crystal packing of **3** did not reveal any notable intermolecular interactions, but revealed that the disordered hexane molecule is located within channels situated along the **c** axis (Figure 5). According to the powder XRD data, due to the loss of solvate molecules, the crystals decay during storage.

### 2.2. Optical Properties

UV-vis spectra of complexes were measured in acetonitrile solutions at room temperature (Appendix A). There are strong absorption bands at the UV region (π→π* electronic transitions) at 400–450 nm and 550–650 nm (MLCT transitions), and less intense absorption in the 650–750 nm range (see Figure 6). The nature of the latter will be discussed in the quantum chemical calculations section. The absorption spectra are decomposed into Gaussian components (Appendix A) to determine the energy of the S_0_→S_1_ transition (λ_abs_^1^) (see Table 2 and Appendix A). For complex **3** in the decomposition into Gaussian components, there is no low-energy maximum similar to the other three complexes. However, in the calculated spectrum there is a transition to 797 nm with a very low oscillator strength. Perhaps its intensity is so low that it is difficult to observe it in the absorption spectrum. For the rest of the complexes, the lowest energy maximum approximately coincides in energy with the theory.

The absorption bands with the maxima at λ_abs_^1^ and λ_abs_^2^ of the complexes shift to longer wavelengths as the donor properties of the ligand increase. This can be explained by the fact that the HOMO of the complex is localized on the benzimidazole ligand; therefore, the introduction of donor substituents increases the HOMO energy, while the introduction of acceptor substituents lowers it.

All the complexes exhibit luminescence in the near-IR region (800–950 nm). The spectra measured at 298 K for compounds **3** and **4** with donor substituents have low-intensity bands at a higher energy region of 600–800 nm. However, these bands are negligible in comparison with the emission bands located at 800–950 nm in the spectra recorded at 77 K. It can be assumed that regarded bands correspond to emission from the singlet state. The lifetime of this state could not be measured because of the low intensity of the band. At room temperature, the emission maxima of complexes **1**–**3** peaked at a wavelength of about 900 nm and slightly blue-shifted in the spectra recorded at 77 K (see Figure 7). Substituents in ligands affect the energy of the emitted excited state with a consequence red shift of the emission band by 81 nm (0.14 eV) (Table 2) from complex **1** to complex **4**. In conclusion, the emission intensity of the long-wavelength band greatly increases upon cooling in comparison with the intensity of the band at 600–800 nm (Appendix A), which indicates the triplet nature of the excited state.

To prove the proposed hypothesis, the luminescence decays at room temperature and 77 K with registration at emission maxima were measured. Decay curves for complexes **1**–**3** are well-fitted by a monoexponential function (see Appendix A), whereas luminescence decay of compound **4** reveals bi-exponential behavior. For the compounds with bi-exponential relaxation of the excited state, the short-time component was employed for further analysis by virtue of significantly higher amplitude (see Appendix A). Excited state lifetimes were estimated as several nanoseconds for the compounds at 298 K. However, at 77 K the lifetimes greatly increase, which also testifies in favor of the assumption about the triplet nature of the excited state. The 392 ± 2 ns lifetime for complex **1** is an unusually long time for fluorescence and instead can be assigned to phosphorescence from T_1_ state. The vibrational relaxation of the T_1_ state energy is fully suppressed at 77 K leading to observable lifetime increase. It should be noted that the lifetime decreases from complex **1** (392 ± 2 ns) to complex **4** (88 ± 1 ns) for both the decays recorded at 77 and 298 K. It is associated with an increase in the donation of substituents.

### 2.3. Quantum Chemical Calculations

To further investigate the influence of the R substituents in the ligand on the electronic properties of the corresponding complexes, quantum chemical calculations were performed for the compounds in the gas phase at the M06 levels of theory with the def2-SV(P) basis set for all atoms except ruthenium. Large-core energy-adjusted quasi-relativistic RECP for Ru, developed by the Stuttgart and Dresden groups, along with the accompanying basis set ECP28MWB, was used. The optimized ground-state geometries for the compounds are in good agreement with the X-ray crystal structures (Appendix A). Cartesian coordinates are given in Appendix A. It should be noted that the phenyl fragment in the Bn unit is twisted by approximately 22°. This difference can be attributed to crystal packing effects. The variation of the substituent units does not significantly change the geometry of the whole complexes, as demonstrated by X-ray single crystal analysis.

The calculated frontier molecular orbital LUMO is predominantly located on the N^N ligand (92%) for all the compounds (see Figure 8). In contrast, the HOMO is presumed to be located on Ru for complexes 1 (67%) and 2 (60%), while for complexes **3** and **4**, the HOMO is located on C^N (61% and 80%, respectively). We observed a trend of decreasing Ru(II) ion contribution in the HOMO along with an increase in C^N ligand contribution from compound **1** to **4** (see Figure 8). The redistribution of molecular orbital localization from the Ru(II) ion to the C^N ligand, specifically on the motif with the R group, leads to an increase in HOMO energy from −8.0 to −7.4 eV.

To gain a deeper insight into electronic excitation processes, TD-DFT calculations were performed at the same level of theory. The calculated S_0_→S_1_ energies agree well with the experimentally estimated values, with an energy deviation not exceeding 0.03 eV (in the case of complex **2**). According to the calculations, the S_0_→S_1_ electronic transition for complexes **1** and **2** has the nature of a metal-to-ligand charge transfer (MLCT) state, involving a transition from Ru to N^N with an additional contribution from a transition from C^N to N^N ligand. Thus, we assume an MLCT (Ru→N^N) + ligand-to-ligand charge transfer (LLCT) (C^N→N^N) nature for this absorption band. However, compounds **3** and **4**, which contain electron-donating substituents, demonstrate an LLCT (C^N→N^N) nature for the S_0_→S_1_ transition. The initial orbital is located on the NMe_2_-containing unit of the C^N ligand.

The first intense absorption band, located in the region of 500–600 nm (see Figure 6), has the nature of an MLCT from the d orbital of Ru to the π* orbital of N^N (Ru→N^N), as determined by calculations. The calculated absorption energies of 577–606 nm do not exceed 0.1 eV compared to those obtained from the deconvolution of experimental spectra. The UV-Vis spectra simulated based on the TD results are quite similar to the measured spectra.

The estimated energies of the first excited triplet state T_1_ are close to the experimental values (Table 3), slightly lower by only 0.05–0.11 eV. Since the S_0_→T_1_ transition is predominantly from HOMO to LUMO, the T_1_ state is an MLCT (Ru→N^N) state. Therefore, it is supported that the introduction of an electron-withdrawing NO_2_ unit increases the T_1_ energy, while the introduction of electron-donating motifs decreases it. 

### 2.4. Electrochemical Studies

CVs obtained for the complexes in a wide range of potentials have the same form (Figure 9). Reversible redox peak at high potentials changes depending on the electron-donating effect of C^N-ligand—the potential lowers with an increase in electron-donating properties. This correlates with the change in HOMO level. The redox peaks at low potentials range practically do not change their position between complexes (Appendix A). This behavior we associate with the electrochemical properties of N^N ligands.

We also estimated the energies of HOMO and LUMO using the oxidation and reduction potentials (Table 4). To do this, we added to the obtained potentials the absolute potential of the pair Fc/Fc^+^ = 5.1 eV [62]. The energy gap (Eg) was calculated as the difference between oxidation and reduction potentials. The dependence of the HOMO and LUMO energies determined in this way on the substituent in the ligand is the same as in the case of quantum chemical calculations. E_LUMO_ for all complexes is approximately the same, except for complex **1**, as in its case, it is lower. This can be explained by the fact that the introduction of a strong acceptor into one ligand leads to a decrease in the electron density on the N^N ligand as well. That is why the E gap is the same for complexes **1** and **2**, although complex **2** does not contain acceptor substituents.

## 3. Materials and Methods

All commercially available reagents were at least reagent grade and used without further purification. Solvents were distilled and dried according to standard procedures. 

NMR spectra were acquired at 25 °C on a Bruker Avance 600 spectrometer and chemical shifts were reported in ppm referenced residual solvent signals.

Single-crystal X-ray diffraction analysis of **3** and **1** was carried out on a Bruker D8 Quest (Bruker, Billerica, MA, USA) diffractometer (MoKα radiation, ω and φ-scan mode). The structure was solved with direct methods and refined by the least squares method in the full-matrix anisotropic approximation on F^2^. All hydrogen atoms were located in calculated positions and refined within the riding model. All calculations were performed using the SHELXTL (version NT) and Olex2 [63,64,65] software packages. Atomic coordinates, bond lengths, angles, and thermal parameters have been deposited at the Cambridge Crystallographic Data Centre with deposition numbers CCDC 2255408 (accesed on 11 April 2023) and 2259495 (accessed on 27 April 2023), which are available free of charge at www.ccdc.cam.ac.uk.

Crystal data for **3** (C_57_H_59_F_6_N_6_O_10_PRu M = 1234.14 g/mol): triclinic; space group P-1 (no. 2); a = 12.930 (3) Å; b = 15.869 (4) Å; c = 16.048 (4) Å; α = 104.525 (7)°; β = 104.525 (7)°; γ = 104.525 (7)°; V = 2760.4 (12) Å^3^; Z = 2; T = 100 (2) K; μ (MoKα) = 0.398 mm^−1^; Dcalc = 1.485 g/cm^3^; 23,695 reflections measured (1.48° ≤ Θ ≤ 25.00°); and 9706 unique (Rint = 0.1010, Rsigma = 0.1288), which were used in all calculations. The final R_1_ was 0.0697 (I > 2σ(I)) and wR_2_ was 0.1932 (all data).

Crystal data for **1** (C_54_H_45_Cl_15_F_6_N_7_O_10_PRu M = 1729.76 g/mol): monoclinic; space group P2_1_/c (no. 14); a = 10.6071 (11) Å; b = 24.277 (3) Å; c = 26.763 (3) Å; β = 100.689 (3)°; V = 6772.2 (12) Å^3^; Z = 4; T = 100 (2) K; μ (MoKα) = 0.924 mm^−1^; Dcalc = 1.697 g/cm^3^; 67,556 reflections measured (1.76° ≤ Θ ≤ 25.00°); and 11,919 unique (Rint = 0.0340, Rsigma = 0.0477), which were used in all calculations. The final R_1_ was 0.0694 (I > 2σ(I)) and wR_2_ was 0.1514 (all data).

Cyclic voltammograms (CVs) of the 5 mM complexes solutions were performed in a standard 3-electrode cell with previously polished 3 mm diameter glassy carbon working electrode, Pt wire counter electrode, and polypyrrole reference electrode. Further, 0.1 M solution of tetrabutylammonium perchlorate (TBAP, Alfa Aesar, electrochemical grade) in Ar-saturated acetonitrile (CH_3_CN) (Sigma-Aldrich, containing 20 ppm H_2_O, established by Fischer titration) was used as an electrolyte. The details of the reference electrode preparation can be found in Ref. [66]. Measurements were performed using BioLogic potentiostat (Seyssinet-Pariset, France) with an analog scan generator at a scan rate of 100 mV s^−1^ at 25 °C. 

MALDI spectra were recorded using a Bruker microTOF II mass spectrometer (Bremen, Germany).

Photoluminescence excitation and emission spectra were recorded at room temperature and 77 K with a Horiba-Jobin-Yvon Fluorolog-QM spectrofluorimeter (Paris, France) equipped with a 75 W ArcTune xenon lamp and a Hamamatsu R13456 photomultiplier sensitive in the 200–980 nm emission range. For low-temperature measurements, the samples were placed in a quartz optical cryostat filled with liquid N_2_. Luminescence decays were acquired by the time-correlated single photon counting (TCSPC) method using the same instrument, using DeltaLED (HORIBA, Kyoto, Japan) as a pulsed laser excitation source emitted at λ = 390 nm and the pulse duration of 0.6 ns. V2.6 software was used for data analysis. To suppress vibrational quenching of phosphorescence, solutions of the complexes were prepared in an argon atmosphere in acetonitrile purified from dissolved oxygen.

Optical absorption spectra were recorded using a LOMO SF-2000 spectrophotometer (Saint Petersburg, Russia) in quartz cuvettes with a 1 cm path length in solutions in acetonitrile.

Quantum chemical calculations were conducted with the Gaussian 16 Rev A.03 program. DFT and TD-DFT calculations were performed at the M06 levels of theory [67,68]. The def2-SV(P) basis set was employed for all atoms except ruthenium. Large-core energy-adjusted quasi-relativistic RECP for Ru, developed by the Stuttgart and Dresden groups, along with the accompanying basis set ECP28MWB [69,70] was used. In the development of the theoretical model, geometrical parameters X-ray single crystal structures were used as a starting point. All calculations were performed in the gas phase. Analysis of vibrational frequencies was performed for all optimized structures. All compounds were characterized by only real vibrational frequencies. The molecular group’s contribution to MOs as well as simulated UV-Vis spectra were calculated using GaussSum 3.0 software [71].

### 3.1. Synthesis of the Ligands


**4-methyl-2-nitro-N-benzylaniline**


The synthesis was carried out according to the method described in [60].

4-methyl-2-nitroaniline (3.77 g, 27.32 mmol), benzylchloride (5 mL, 43.45 mmol, 1.6 eq.), KBr (6 g), and water (55 mL) were added to a 250 mL round-bottomed flask equipped with a reverse refrigerator and an anchor of a magnetic stirrer; the reaction mixture was stirred for 1.5 h at boiling, and then the solution was cooled to room temperature, treated with a saturated solution of sodium bicarbonate (2.45 g), extracted with ethyl acetate (2 × 65 mL), and washed with water (50 mL). The organic phase was evaporated on a rotary evaporator and crystallized from ethanol (Figure 1). The yield was: 4.73 g (72%) in the form of red-orange crystals. ^1^H NMR (CDCl_3_, 600 MHz) δ: 8.33 (s, 1H), 7.99 (s, 1H), 7.38–7.19 (m, 6H), 6.72 (d, 1H), 4.53 (d, 2H), and 2.25 (s, 3H).


**N-benzyl-4-methylphenylendiamine**


The synthesis was carried out according to the method described in [59].

4-methyl-2-nitro-N-benzylaniline (1.21 g, 5 mmol), obtained Raney nickel, and methanol (24 mL) were added to a round-bottomed flask with a volume of 100 mL, equipped with a reverse refrigerator and an anchor of a magnetic stirrer. Then, hydrazine hydrate (1.25 g, 5 mmol) was added drop by drop. The reaction mixture was stirred for 3–5 h at 70 °C. Excess hydrazine was removed by adding another portion of nickel Raney. The resulting solution was purified from nickel by column chromatography and evaporated on a rotary evaporator (Figure 1). The yield was 0.8 g, (76%). The target compound was obtained in the form of yellow oil. The obtained substances were used in the next step without characterization because they oxidize rapidly.


**General method of obtaining bisulfite adducts**


Bisulfite adducts were obtained by the method [72].

The corresponding aldehyde was dissolved in a minimum amount of ethanol. A saturated aqueous solution of sodium pyrosulfite was added to the solution with stirring. The resulting suspension was stirred for several minutes, the precipitate was filtered out and washed 2–3 times with cold alcohol. The white powder was dried in a desiccator over phosphorus oxide (V). The outputs are practically quantitative.


**General method of synthesis of 2-aryl-N-benzyl-5-methylbenzimidazoles**


The synthesis was carried out according to the method in [58].

4-methyl-N-benzylphenylenediamine (1 g, 4.75 mmol) dissolved in ethanol (5 mL) and a bisulfite adduct of the corresponding aldehyde (5.7 mmol, 1.2 eq.) dissolved in ethanol (10 mL) were added to a 50 mL round-bottomed flask equipped with a reverse refrigerator and an anchor of a magnetic stirrer. The reaction mixture was stirred for 3–5 h during boiling. At the end of this time, the reaction mixture was cooled and filtered at reduced pressure, and the inorganic phase was removed from the precipitate by washing it with water. Next, the target product was washed off the filter with hot ethanol, the product that fell out of the saturated solution was filtered, and the remainder was crystallized from an aqueous alcohol solution (Figure 1). NMR spectra are given in Appendix A.

**L-NO_2_** (1-benzyl-2-(4-nitrophenyl)benzimidazole) pale-yellow powder, yield 73%. 

^1^H NMR (600 MHz, Chloroform-d) δ 8.31 (d, *J* = 8.5 Hz, 2H), 7.90 (d, *J* = 8.5 Hz, 2H), 7.72 (s, 1H), 7.35 (dq, *J* = 13.7, 6.7 Hz, 3H), 7.18 (t, *J* = 6.1 Hz, 2H), 7.09 (d, *J* = 6.9 Hz, 2H), 5.48 (s, 2H), and 2.52 (s, 3H).

**L-H** (1-benzyl-2-phenylbenzimidazole) white powder, yield 75%. 

^1^H NMR (600 MHz, Chloroform-d) δ 7.61 (s, 1H), 7.57 (d, *J* = 8.9 Hz, 2H), 7.33 (t, *J* = 7.3 Hz, 2H), 7.29 (t, *J* = 7.3 Hz, 1H), 7.14 (d, *J* = 7.0 Hz, 2H), 7.05–6.98 (m, 2H), 6.72 (d, *J* = 8.9 Hz, 2H), 5.43 (s, 2H), and 2.48 (s, 3H).

**L-OMe_2_** (1-benzyl-2-(3,4-dimethoxyphenyl)benzimidazole) white powder, yield 71%. 

^1^H NMR (600 MHz, Chloroform-d) δ 7.64 (s, 1H), 7.34 (t, *J* = 7.3 Hz, 2H), 7.29 (t, *J* = 7.3 Hz, 1H), 7.22–7.19 (m, 2H), 7.12 (d, *J* = 7.2 Hz, 2H), 7.11–7.04 (m, 2H), 6.90 (d, *J* = 8.1 Hz, 1H), 5.43 (s, 2H), 3.91 (s, 3H), 3.71 (s, 3H), and 2.50 (s, 3H).

**L-NMe_2_** (1-benzyl-2-(4-dimethylaminophenyl)benzimidazole) white powder, yield 77%. 

^1^H NMR (600 MHz, Chloroform-d) δ 7.65 (s, 1H), 7.59 (d, *J* = 8.9 Hz, 2H), 7.34 (t, *J* = 7.3 Hz, 2H), 7.30 (t, *J* = 7.3 Hz, 1H), 7.14 (d, *J* = 7.1 Hz, 2H), 7.03 (t, *J* = 7.0 Hz, 2H), 6.72 (d, *J* = 8.9 Hz, 2H), 5.44 (s, 2H), 3.00 (s, 6H), and 2.48 (s, 3H).

### 3.2. Synthesis of the Complexes ***1***–***4***

Complexes **1**–**4** were synthesized in two steps according to a general procedure:

(1) [Ru(p-cymene)Cl]_2_Cl_2_ (153.0 mg, 0.25 mmol) 1-benzyl-2-aryl-5-methylbenzimidazole (0.5 mmol), NaOH (20.0 mg, 0.5 mol), KPF_6_ (184.0 mg, 1 mmol) were dissolved in 5 mL of anhydrous acetonitrile in a vial, carefully degassed with Ar, sealed, and kept at 45 °C for 72 h. The solution turned from orange to dark yellow or greenish yellow. The resulting mixture was evaporated and purified by column chromatography (silica gel, eluent CH_2_Cl_2_ -> CH_3_CN:CH_2_Cl_2_ (1:10 vol.)). The bright yellow fraction was isolated, reconstituted in 5 mL of dichloromethane, and precipitated using 20 mL of hexane to afford a yellow solid, which was used in the next step without further purification. 

(2) [RuL(CH_3_CN)_4_]PF_6_ (1 eq.) and 4,4′-dicarboxylic-2,2′-bipyridine dimethyl ester (1.8 eq.) were refluxed in a mixture of methanol:dichloromethane (1:1 vol.) for 5 h under argon. The mixture almost immediately changed color from yellow to intense black-green or black-violet. The resulting mixture was purified by column chromatography (silica gel, eluent CH_2_Cl_2_:CH_3_OH (20:1 vol.)). The dark band was collected and recrystallized from a dichloromethane–methanol–hexane mixture. The yields were calculated relative to [Ru(p-cymene)Cl]_2_Cl_2_. ^1^H, ^13^C, and COSY NMR spectra and assignment of protons are presented in SI (Appendix A). For complex **3**, the HMBC spectrum was registered, and the assignment of carbons in ^13^C was made.

**[Ru(L-NO_2_)(dmdcbp)_2_]PF_6_ (1):** dark violet powder, yield 51.5%, 29.2 mg.

UV-vis (most intensive bands): 571 nm, 420 nm.

MALDI *m*/*z*: [M]^+^ Calcd for C_49_H_40_N_7_O_10_Ru^+^ 988.1881; found 988.1888.

^1^H NMR (600 MHz, Acetone-*d*_6_) δ 9.32 (s, 1H), 9.21–9.18 (m, 1H), 9.17–9.15 (m, 1H), 9.15–9.11 (m, 1H), 8.55 (d, *J* = 5.3 Hz, 1H), 8.35 (dd, *J* = 5.9, 1.6 Hz, 2H), 8.28 (d, *J* = 6.0 Hz, 1H), 8.10 (dd, *J* = 5.7, 1.6 Hz, 1H), 8.09 (d, *J* = 8.7 Hz, 1H), 7.86 (d, *J* = 4.2 Hz, 1H), 7.85–7.84 (m, 1H), 7.84–7.82 (m, 1H), 7.66–7.63 (m, 1H), 7.63–7.61 (m, 1H), 7.34 (t, *J* = 7.2 Hz, 2H), 7.30 (d, *J* = 7.0 Hz, 1H), 7.16 (d, *J* = 7.8 Hz, 1H), 7.15–7.11 (m, 3H), 6.23–6.13 (m, 2H), 5.69 (s, 1H), 4.03 (s, 3H), 3.99 (s, 3H), 3.98–3.95 (m, 6H), and 2.04 (s, 3H).

^13^C NMR (151 MHz, Acetone) δ 194.64, 165.21, 165.08, 165.01, 164.97, 159.05, 158.89, 157.93, 157.58, 156.15, 153.08, 152.71, 152.23, 147.53, 143.84, 142.17, 138.99, 138.06, 137.21, 136.94, 136.69, 136.59, 136.37, 136.20, 134.93, 130.01, 130.01, 128.98, 128.84, 127.06, 126.97, 126.85, 126.85, 126.75, 126.65, 126.41, 124.04, 124.04, 123.94, 123.88, 117.95, 115.67, 112.11, 53.69, 53.52, 53.52, 53.45, 49.01, and 21.45.

**[Ru(L-H)(dmdcbp)_2_]PF_6_ (2):** dark violet powder, yield 64.8%, 35.3 mg.

UV-vis (most intensive bands): 592 nm, 426 nm.

MALDI *m*/*z* [M]^+^ Calcd for C_49_H_41_N_6_O_8_Ru^+^ 943.2030; found 943.2037. 

^1^H NMR (600 MHz, Acetone-*d*_6_) δ 9.29 (d, *J* = 0.9 Hz, 1H), 9.13 (d, *J* = 1.3 Hz, 1H), 9.10 (d, *J* = 1.7 Hz, 2H), 8.53 (dd, *J* = 5.7, 0.7 Hz, 1H), 8.39 (dd, *J* = 6.0, 0.6 Hz, 1H), 8.36 (dd, *J* = 5.9, 0.6 Hz, 1H), 8.22 (dd, *J* = 6.0, 0.6 Hz, 1H), 8.09 (dd, *J* = 5.6, 1.6 Hz, 1H), 7.86 (dd, *J* = 7.8, 1.1 Hz, 1H), 7.84 (dd, *J* = 6.0, 1.8 Hz, 1H), 7.83 (dd, *J* = 6.0, 1.8 Hz, 1H), 7.79 (dd, *J* = 6.0, 1.8 Hz, 1H), 7.54–7.50 (m, 1H), 7.34–7.30 (m, 2H), 7.30–7.26 (m, 1H), 7.13–7.10 (m, 2H), 7.06 (dd, *J* = 8.4, 1.0 Hz, 1H), 6.84 (td, *J* = 7.5, 1.5 Hz, 1H), 6.80 (td, *J* = 7.3, 1.4 Hz, 1H), 6.35 (dd, *J* = 7.3, 1.2 Hz, 1H), 6.09 (d, *J* = 3.2 Hz, 2H), 5.61 (s, 1H), 4.02 (s, 3H), 3.98–3.96 (m, 9H), and 2.02 (s, 3H).

^13^C NMR (151 MHz, Acetone-*d*_6_) δ 191.42, 165.25, 165.17, 165.07, 165.07, 160.73, 159.09, 159.09, 157.87, 157.58, 155.72, 152.81, 152.39, 152.11, 152.02, 142.09, 138.56, 137.07, 137.00, 136.68, 135.92, 135.92, 135.81, 135.41, 134.16, 129.89, 129.70, 128.63, 126.85, 126.85, 126.69, 126.51, 126.39, 126.28, 125.75, 123.89, 123.76, 123.67, 123.60, 123.02, 115.16, 111.57, 66.12, 53.64, 53.45, 53.40, 53.40, 48.84, and 21.42.

**[Ru(L-OMe_2_)(dmdcbp)_2_]PF_6_ (3):** dark green powder, yield 69.7%, 40.0 mg.

UV-vis (most intensive bands): 597 nm, 422 nm.

MALDI *m*/*z*: [M]^+^ Calcd for C_51_H_45_N_6_O_10_Ru^+^ 1003.2241; found 1003.2259.

^1^H NMR (600 MHz, Acetone-*d*_6_) δ 9.28 (s, 1H), 9.13 (d, *J* = 1.3 Hz, 1H), 9.10 (d, *J* = 1.3 Hz, 1H), 9.08 (d, *J* = 1.3 Hz, 1H), 8.54 (d, *J* = 5.6 Hz, 1H), 8.43 (d, *J* = 5.9 Hz, 1H), 8.41 (d, *J* = 5.9 Hz, 1H), 8.26 (d, *J* = 6.0 Hz, 1H), 8.08 (dd, *J* = 5.6, 1.5 Hz, 1H), 7.88–7.84 (m, 2H), 7.80 (dd, *J* = 5.9, 1.7 Hz, 1H), 7.51 (d, *J* = 8.4 Hz, 1H), 7.34 (t, *J* = 7.3 Hz, 2H), 7.32 (s, 1H), 7.29 (t, *J* = 7.3 Hz, 1H), 7.16–7.11 (m, 2H), 7.04–7.00 (m, 1H), 6.09–6.01 (m, 2H), 5.80 (s, 1H), 5.54 (s, 1H), 4.03 (s, 3H), 3.97 (s, 6H), 3.96 (s, 3H), 3.51 (s, 3H), 3.32 (s, 3H), and 2.02 (s, 3H).

^13^C NMR (151 MHz, Acetone-*d*_6_) δ 183.67, 165.30, 165.16, 165.12, 165.12, 161.02, 159.31, 159.06, 157.85, 157.74, 155.93, 152.85, 152.44, 152.12, 151.28, 146.37, 142.16, 138.45, 137.32, 136.54, 135.94, 135.57, 135.13, 133.81, 129.96, 129.96, 128.64, 127.63, 126.89, 126.67, 126.67, 126.56, 126.35, 126.22, 125.01, 123.82, 123.63, 123.57, 123.56, 117.31, 114.59, 111.26, 111.04, 56.08, 55.17, 53.64, 53.44, 53.39, 53.35, 48.62, and21.41.

**[Ru(L-NMe_2_)(dmdcbp)_2_]PF_6_ (4):** dark green powder, yield 69.3%, 39.2 mg.

UV-vis (most intensive bands): 603 nm, 395 nm.

MALDI *m*/*z*: [M]^+^ Calcd for C_51_H_46_N_7_O_8_Ru^+^ 986.2452; found 986.2460. 

^1^H NMR (600 MHz, Acetone-*d*_6_) δ 9.27 (d, *J* = 1.2 Hz, 1H), 9.12 (d, 1H), 9.09 (d, 2H), 8.54 (dd, 1H), 8.51 (dd, 1H), 8.43 (dd, 1H), 8.26 (dd, 1H), 8.07 (dd, *J* = 5.7, 1.7 Hz, 1H), 7.87–7.83 (m, 2H), 7.79 (dd, *J* = 6.0, 1.8 Hz, 1H), 7.66 (d, *J* = 8.8 Hz, 1H), 7.38 (d, *J* = 8.3 Hz, 1H), 7.31 (t, *J* = 7.1 Hz, 2H), 7.27 (t, *J* = 7.3 Hz, 1H), 7.10 (d, *J* = 7.2 Hz, 2H), 6.93 (d, *J* = 8.0 Hz, 1H), 6.20 (dd, *J* = 8.9, 2.6 Hz, 1H), 5.95 (s, 2H), 5.56 (d, *J* = 2.6 Hz, 1H), 5.48 (s, 1H), 4.02 (s, 3H), 3.98–3.97 (m, 6H), 3.96 (s, 3H), 2.64 (s, 6H), and 1.99 (s, 3H).

^13^C NMR (151 MHz, Acetone-*d*_6_) δ 192.22, 165.37, 165.24, 165.17, 165.17, 161.42, 159.47, 159.09, 157.87, 157.64, 155.71, 152.80, 152.51, 152.11, 150.84, 142.39, 138.39, 137.30, 136.48, 135.86, 135.35, 135.11, 133.45, 129.86, 129.86, 128.55, 127.36, 127.30, 126.93, 126.93, 126.40, 126.28, 126.14, 124.38, 123.82, 123.62, 123.58, 123.40, 117.63, 114.33, 112.91, 110.82, 107.66, 53.66, 53.43, 53.43, 53.20, 48.50, 39.65, 39.65, and 21.45.

## 4. Conclusions

In conclusion, this article presents a complete study of a series of cyclometalated ruthenium 2-arylbenzimidazole complexes. The substituents in the C^N ligand play an important role in the electronic structure of the complex. It has been shown that an increase in the donation of a substituent in the aryl fragment of benzimidazole leads to a red shift of the emission band and a decrease in the lifetime of the excited state; an assumption was also made about the triplet nature of the excited state. In addition, an increase in donation shifts the absorption band to the red region of the spectrum, but, as we showed using quantum chemical calculations, the contribution of interligand energy transfer also increases. At the same time, the introduction of a strong acceptor affects even the energies of the orbitals localized on the N^N ligand. The results obtained are important for a deep understanding of the nature of electronic transitions and the targeted development of effective ruthenium dyes for solving various problems.

## Data Availability

Data are contained within the article and Appendix A.

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
