# Peer review of "Fine-Tuning of the Optical and Electrochemical Properties of Ruthenium(II) Complexes with 2-Arylbenzimidazoles and 4,4′-Dimethoxycarbonyl-2,2′-bipyridine"

_molecules, 2023, doi:10.3390/molecules28186541_

Round 1
Reviewer 1 Report
The manuscript reports a preparation of four organic molecules and successful synthesis of four complexes of Ru(II) cations with those ligands. The complexes were characterized by a single-crystal X-ray diffraction, spectroscopic methods and mass-spectrometry. The photoluminescence properties as well as electrochemical behavior were also investigated and rationalized by quantum chemical calculations. The experiments are carried out well, the discussion is clear and consistent, the conclusions are based on the available experimental evidences and literature data. This paper could be accepted for publication although some minor details are to be revised first.
1) Notations “N^N” or “N^C” (introduction section) are not common and require some explanation. A scheme might be helpful.
2) Scheme 1. The reagent above the arrow in the last step is wrong. It is a sulfonate, not an aldehyde adduct.
3) Page 3, lines 122-124 refer to changes in NMR spectra of N^N ligand. It would be helpful to add the spectrum of the bpy molecule to the Figure 1.
4) Page 5, lines 151-152. Please, specify a version and/or the actual data of the Cambridge Database used for the search and reference.
5) Page 7, lines 200-202. The increase/decrease of the intensities are not apparent from the Fig.6 as all the spectra are normalized. Please, report quantum yields or provide the emission spectra at different temperatures on one plot with a common scale.
6) Page 13. Line 401. Is the metal to ligand ratio really 1:0.9? The composition of the final complex is 1:2, therefore, an excess of the bpy ligand is suggested. Please, check. Also, the chemical yields (mg and percentage) of the final complexes 1-4 should be reported.
7) Please, elaborate the role of the carboxyester substituents on the bpy ligands of the Ru(II) complexes. Why not use 4,4’-bpy? What is an influence of those ester functions on optical, electrochemical properties or solubility of the complexes?
8) Figure 1. Please, revise Cyrillic characters.
9) Figures 2, 3. Please, provide a legend to the colors of the atoms.
Author Response
Reviewer 1
Dear Reviewer,
We are grateful to you for reading our article and valuable comments, we tried to take them all into account.
1) Notations “N^N” or “N^C” (introduction section) are not common and require some explanation. A scheme might be helpful.
Answer: we've added a transcription of these notations to the introduction.
2) Scheme 1. The reagent above the arrow in the last step is wrong. It is a sulfonate, not an aldehyde adduct.
Answer: We have corrected the scheme.
3) Page 3, lines 122-124 refer to changes in NMR spectra of N^N ligand. It would be helpful to add the spectrum of the bpy molecule to the Figure 1.
Answer: The spectrum of the ligand molecule is already shown in the figure (top).
4) Page 5, lines 151-152. Please, specify a version and/or the actual data of the Cambridge Database used for the search and reference.
Answer: We have specified the version of CSD.
5) Page 7, lines 200-202. The increase/decrease of the intensities are not apparent from the Fig.6 as all the spectra are normalized. Please, report quantum yields or provide the emission spectra at different temperatures on one plot with a common scale.
Answer: Accurate measurement of quantum luminescence yields at the boundary of the infrared and visible spectral regions is a difficult task. For this reason, normalized spectra measured at different temperatures are presented in the SI.
6) Page 13. Line 401. Is the metal to ligand ratio really 1:0.9? The composition of the final complex is 1:2, therefore, an excess of the bpy ligand is suggested. Please, check. Also, the chemical yields (mg and percentage) of the final complexes 1-4 should be reported.
Answer: We have corrected the experimental pert; we take 1:1.8 ratio of [RuL(CH3CN)4]PF6 and dmdcbp, because [RuL(CH3CN)4]PF6 was not purified from residual p-cymene, and when ratio 1:2 is taken, it is hard to separate the final product from an excess of dmdcbp. We have added the explanation in «2.1. Synthesis and characterization», lines 146-149. We have also added the yields in mg.
7) Please, elaborate the role of the carboxyester substituents on the bpy ligands of the Ru(II) complexes. Why not use 4,4’-bpy? What is an influence of those ester functions on optical, electrochemical properties or solubility of the complexes?
Answer: We have added an explanation of the choice of N^N ligand in the introduction.
8) Figure 1. Please, revise Cyrillic characters.
Answer: We`ve corrected the caption.
9) Figures 2, 3. Please, provide a legend to the colors of the atoms.
Answer: We have added the legend.

Reviewer 2 Report
Dear authors, I recommend the manuscript for publication. However, there are some remarks.
1. In the second section of the supplementary information, in addition to the figures, the Cartesian coordinates of the atoms of the DFT optimized ground-state structure should be given.
2. The article would be greatly embellished if in Table 1, along with the X-ray experimental bond lengths, the DFT calculated values would also be given.
3. In modern publications, in addition to 1H NMR spectra, 13C NMR spectra are also given. Therefore, in the experimental part (or in the supplement), the 13C NMR spectra for the proform should also be given.
4. Also in the experimental part, in addition to the CCDC numbers, the lattice parameters and the space group of the crystals should be given as documentary material.
Author Response
Reviewer 2
Dear Reviewer,
We are grateful to you for reading our article and valuable comments, we tried to take them all into account.
- In the second section of the supplementary information, in addition to the figures, the Cartesian coordinates of the atoms of the DFT optimized ground-state structure should be given.
Answer: We have added the Cartesian coordinates in supplementary information.
- The article would be greatly embellished if in Table 1, along with the X-ray experimental bond lengths, the DFT calculated values would also be given.
Answer: We have added table with comparison of X-ray experimental bond lengths and the DFT calculated values to supplementary information.
- In modern publications, in addition to 1H NMR spectra, 13C NMR spectra are also given.Therefore, in the experimental part (or in the supplement), the 13C NMR spectra for the proform should also be given.
Answer: We have added 13C and COSY spectra to supplementary information, as well as signal assignment in 1H spectrum.
- Also in the experimental part, in addition to the CCDC numbers, the lattice parameters and the space group of the crystals should be given as documentary material.
Answer: We have added crystal data in experimental part.

Reviewer 3 Report

The text should be thoroughly checked.
Author Response
Reviewer 3
Dear Reviewer,
We are grateful to you for reading our article and valuable comments, we tried to take them all into account. All the minor pointes were corrected.
Major points
1) The word dye within the text is wrongly used. Following the literature references I realized that they refer to molecules sensitizers for Dye sensitized solar cells. This should be clearly written in the text. This is not the case herein.
Introduction part may be changed, including a part that clearly refers to DSCC’s. Some references may be added therein: Coord. Chem. Rev. 2011, 255, 2602-2621; Dalton Trans. 51 (2022) 15049-15066 etc.
Answer: thanks for the comment, corrected the use of the word dye on the text, and expanded the reference list.
2) In the Introduction part as well, a Scheme must be introduced to summarize the situation with previously used complexes and similar ligands.
Answer: We have added a scheme illustrating the types of ruthenium compounds with benzimidazoles described in the literature and enlarged the part in the introduction devoted to complexes with benzimidazoles.
3) The experimental part is inappropriately presented. The synthesis of each complex should contain all experimental data in the following way. Yield, UV-vis data, 1H NMR, elemental analyses and NOT only the ESI-MS data. This should be in accord to the instruction of authors from the journal.
Answer: We have added experimental data (colour, yield, UV-vis data, ESI-MS, 1H, 13C).
4) Elemental analyses are missing in the experimental. Although ESI-MS data support the structures proposed, it is not clear if these refer to the bulk or only to the crystalline material isolated.
Answer: In our case, elemental analysis is not informative because the structure contains a non-stoichiometric amount of solvents. Mass spectrometry data refer to bulk material. To refine the NMR spectroscopy data, we made COSY NMR and correlated the proton signals.
5) Line 110. Use of ester moiety vs -COOH one is not very convincing, since it is well-known that -COOH moiety is more appropriate as it works better as a dye in DSCC’s.
Answer: We have added an explanation of the choice of N^N ligand in the introduction.
6) References must be written in a correct way. Check for example references 4, 26, 30,36, 49.
Answer: We have corrected references.

Round 2
Reviewer 3 Report
The authors have taken my comments into consideration.
Thus, I am glad to suggest acceptance in its current form
Minor editing is required